# Anti-Inflammatory Chilean Endemic Plants

**DOI:** 10.3390/pharmaceutics15030897

**Published:** 2023-03-10

**Authors:** Carolina Otero, Carolina Klagges, Bernardo Morales, Paula Sotomayor, Jorge Escobar, Juan A. Fuentes, Adrian A. Moreno, Felipe M. Llancalahuen, Ramiro Arratia-Perez, Felipe Gordillo-Fuenzalida, Michelle Herrera, Jose L. Martínez, Maité Rodríguez-Díaz

**Affiliations:** 1Escuela de Química y Farmacia, Facultad de Medicina, Universidad Andrés Bello, Santiago 8320000, Chile; 2Instituto de Investigación Interdisciplinar en Ciencias Biomédicas SEK, Facultad de Ciencias de la Salud, Universidad SEK, Santiago 8320000, Chile; 3Departamento de Biología, Facultad de Química y Biología, Universidad de Santiago de Chile, Santiago 9160000, Chile; 4Departamento de Urología, Facultad de Medicina, Pontificia Universidad Católica de Chile, Santiago 8320000, Chile; 5Laboratorio de Química Biológica, Facultad de Ciencias, Pontificia Universidad Católica de Valparaíso, Valparaíso 2340000, Chile; 6Laboratorio de Genética y Patogénesis Bacteriana, Facultad de Ciencias de la Vida, Universidad Andrés Bello, Santiago 8320000, Chile; 7Centro de Biotecnología Vegetal, Facultad de Ciencias de la Vida, Universidad Andrés Bello, Santiago 8320000, Chile; 8Laboratorio de Fisiopatología Integrativa, Facultad de Ciencias de la Vida, Universidad Andrés Bello, Santiago 8320000, Chile; 9Center for Applied Nanoscience, Universidad Andrés Bello, Santiago 8320000, Chile; 10Laboratorio de Microbiología Aplicada, Centro de Biotecnología de los Recursos Naturales, Facultad de Ciencias Agrarias y Forestales, Universidad Católica del Maule, Talca 3460000, Chile; 11Vicerrectoria de Investigación, Desarrollo e Innovación, Universidad de Santiago de Chile, Santiago 9160000, Chile; 12Facultad de Ciencias Biológicas, Universidad Nacional de Trujillo, Trujillo 13001, Peru; 13Facultad de Farmacia y Bioquímica, Universidad Nacional de Trujillo, Trujillo 13001, Peru

**Keywords:** bioactive compounds, Chilean native plants, anti-inflammatory properties, phytochemicals, medicinal plants

## Abstract

Medicinal plants have been used since prehistoric times and continue to treat several diseases as a fundamental part of the healing process. Inflammation is a condition characterized by redness, pain, and swelling. This process is a hard response by living tissue to any injury. Furthermore, inflammation is produced by various diseases such as rheumatic and immune-mediated conditions, cancer, cardiovascular diseases, obesity, and diabetes. Hence, anti-inflammatory-based treatments could emerge as a novel and exciting approach to treating these diseases. Medicinal plants and their secondary metabolites are known for their anti-inflammatory properties, and this review introduces various native Chilean plants whose anti-inflammatory effects have been evaluated in experimental studies. *Fragaria chiloensis*, *Ugni molinae*, *Buddleja globosa*, *Aristotelia chilensis*, *Berberis microphylla*, and *Quillaja saponaria* are some native species analyzed in this review. Since inflammation treatment is not a one-dimensional solution, this review seeks a multidimensional therapeutic approach to inflammation with plant extracts based on scientific and ancestral knowledge.

## 1. Introduction

Inflammation is a natural defense of the body against threatening stimuli such as allergens or injuries [1]. Studies have elucidated that inflammation is crucial in different disorders such as cancer, cardiovascular diseases, diabetes, eye disorders, arthritis, autoimmune diseases, obesity, and inflammatory bowel disease. However, which factors are responsible for inflammation? Free radical production results from a lack of natural antioxidants, which might trigger an inflammatory environment. Investigations of inflammatory-related diseases have mostly clarified the fundamental role of free radicals in cellular and tissue damage [2].

Normal cellular metabolism produces reactive oxygen species (ROS) (e.g., superoxide anion, hydrogen peroxide, hydroxyl radical, and organic peroxides), which play a crucial role in cell activation, modifying intra- and extracellular metabolism. Most ROS are produced through the mitochondrial respiratory chain, where nucleic acids, lipids, and proteins are essential cellular components targeted by the oxidative attack. Consequently, modifications in these biomolecules can lead to cell malfunction or even an increased mutagenesis rate [3].

In an inflammatory response, cells generate soluble inflammatory mediators such as cytokines, arachidonic acid, and chemokines, which act through other active inflammatory cells in the infection area, releasing more reactive species. These important markers can trigger signal transduction cascades with alterations of transcription factors, such as nuclear factor kappa B (NF-κB). In addition, the initiation of cyclooxygenase-2 (COX-2), the inducibility of nitric oxide synthase (iNOS), and high expression of inflammatory cytokines, including tumor necrosis factor-α (TNF-α), interleukin-1β (IL-1β), IL-6, and several chemokines, also have a role in oxidative stress-induced inflammation. This oxidative environment provokes an inflammatory condition resulting in an unhealthy circle, damaging normal cells and healthy epithelium [4].

There are several drugs for regulating and suppressing an inflammatory crisis, such as steroidal and nonsteroidal drugs (NSAIDs) and immuno-suppressants [5]. However, these drugs are associated with adverse effects [5]. For example, NSAIDs have well-known adverse effects affecting the gastric mucosa and renal, cardiovascular, hepatic, and hematologic systems [5]. Because these adverse effects of many anti- inflammatory drugs occur at a much higher rate in patients with specific comorbidities, it is important for health professionals to pay close attention to a patient’s history and to educate the patient accordingly on risks and dosing. Therefore, the primary strategy is using minimum effective dosages with the highest efficacy and minor adverse effects. To this end, natural anti-inflammatory factors have been progressively gaining attention since these factors have been shown to produce the lowest degree of undesirable side effects and an excellent pharmacological response [6]. In this sense, Chile is one of the planet’s top five plant biodiversity hotspots.

In this review, we assessed several native Chilean plants (Table 1) with pharmacological evidence of their anti-inflammatory effects, through in vivo, in vitro, and/or clinical studies. These plants, including several edible plants, could offer alternative treatments for various diseases of global interest because of their anti-inflammatory traits.

## 2. Native Plants and Their Metabolites

Since prehistoric times, humans have observed animals’ instinctive behavior when they have to heal their wounds or alleviate their diseases. They have also learned to distinguish between edible and toxic species and identify plants with medicinal effects. In America, the first indicators of the use of medicinal plants are located in Monteverde, an archaeological site 36 km from Puerto Montt (Chile). Researchers found boldo traces there, albeit boldo was a non-endemic species, suggesting that the first inhabitants gave a therapeutic value to this plant. In Chile, before the arrival of the Spanish sailors, the Mapuches used medicinal herbs within their treatments and were aware of more than 200 plants with therapeutic properties. The first apothecary was found in Santiago, where the Mapuches manufactured numerous medicinal plant preparations. Subsequently, a considerable percentage of these indigenous Chilean plants were prepared in 1772 by Brother José Zeitler, who stayed in the pharmacy after the expulsion of the Jesuits.

Today, the most common use of medicinal plants in Chile is recognized to involve the native sources and feral species brought by Europeans [40,41]. The Chilean Ministry of Health has published a book that includes 103 species of native and introduced plants, which have permission to be marketed in the country as traditional herbal medicines or as phytotherapeutic preparations [42]. The current health legislation in Chile considers that pharmaceuticals prepared with plant ingredients can be included in different categories on what was implemented by the public health institute in the supreme decree (DS 3/2010 of medicines and DS 977/96 of food) [42].

The presence of different cultures and indigenous peoples in Chile have given more profound value to the botanical arsenal inhabiting the region from the Atacama Desert to Patagonia since plants were not only used for medicinal purposes but also religious, ornamental, and craft applications. Some people have left a rich cultural legacy concerning traditional medicine and are commonly recognized in Chilean history, including Kawésqar, Mapuche, Picunche, and Aymara, among others. These people used herbal medicine through ancestral knowledge or empirical medical practice [43].

Before describing how modern studies have contributed to the knowledge of the plants used by the indigenous settlements, the corresponding scientific validation of their medicinal properties should be mentioned, as well as the compounds responsible for these properties. All plants are characterized by their different phytochemical profiles, which are responsible for primary metabolism (indispensable compounds for plant life) and secondary metabolism; these compounds are responsible for color, flavor, and bioactive properties according to environmental factors, including water availability, temperature, climate, salt, and cultivation period [44]. In addition, some secondary metabolic compounds have been reported to contribute to a plant’s antioxidant, antibacterial, and anti-inflammatory properties [45,46,47]). Due to their bioactive utility, many indigenous Chilean plants have been used for their medicinal properties and other purposes, such as in health foods, anti-obesity medications, antioxidants, anticancer agents, and cosmetics [48,49,50,51].

Bioactive compounds can be found in fresh and dried plant material, including leaves, stems, roots, seeds, and fruits [52,53]. Many active ingredients are simple to detect and isolate, while others, as part of complex mixtures, are difficult to analyze and therefore determine the active compound, as is the case with many essential oils or resinous substances [54,55,56,57,58]. An example of the difficulty in extracting and isolating the active ingredients from a complex extract is purifying desacylsaponins and obtaining pentacyclic triterpenes such as quillaic acid from quillai saponins [54].

The different chemical components of plants are generally abundant and display diverse structures. Primary metabolites are found in all plants and play vital functions in obtaining energy, morphogenesis, and reproduction. Secondary metabolites are considered non-essential for life, although they may be fundamental for specific biological functions. They are compounds of great pharmacological interest and the design basis for functional foods or supplements [59,60]. Secondary metabolites are grouped into terpenoids, flavonoids, phenolic compounds, and alkaloids, among other classifications, according to the type of genin or aglycone [61]. These structurally variable phytochemical compounds have been studied for their anti-inflammatory properties in the species mentioned in this review. Some of these significant bioactive phytochemicals are shown in Figure 1.

For secondary metabolites research, it is necessary to extract these from plant material. For this, vegetal extract can be processed fresh or dry and duly fragmented, considering the moment of collection of the vegetal material [62]. The study of the plant material starts with pre-extraction and extraction procedures, which are essential steps in processing bioactive constituents from plant materials [62]. Traditional methods, such as maceration and Soxhlet extraction, are commonly used in small research settings [63]. However, significant advances have been made in processing medicinal plants using modern extraction methods; microwave-assisted, ultrasound-assisted, and supercritical fluid extraction methods aim to increase yields at a lower cost [64,65]. Moreover, modifications in purification methods are continuously being developed. With such a variety of different methods, the selection of a proper extraction method needs accurate evaluation [64].

Recently, metabolomics has been used as a powerful tool for metabolite analysis in food quality control, microorganism’s metabolism, and comprehensive identification and quantification of metabolites in plants [66]. Therefore, metabolomics based on liquid chromatographic separation combined with mass spectrometry offers detailed information on plant metabolites and could be advantageous in botanical chemotaxonomy [67]. Furthermore, beneficial secondary compounds in plants, such as flavonoids, phenolic compounds, and terpenoids, have been identified, along with plant bioactivities, including antioxidant activity, anti-microbial activity [38,68,69,70], tyrosinase inhibition activity [71,72], and anti-inflammatory activity [18,29,34,73,74]. However, few studies have attempted to demonstrate a relationship between metabolite differences and bioactivity in diverse native Chilean plants. In this review, we have endeavored to assess several native Chilean plants and the clinical evidence of their anti-inflammatory effects.

### 2.1. Ugni molinae Turcz (murta)

*Ugni molinae* Turcz is commonly known as murta, murtilla, myrtle berry, mutilla, or Chilean guava (Figure 2) [75]. Murta is a small evergreen shrub that grows in the coastal and Andes Mountain ranges in the south-central region of Chile. During summer, murta can be eaten as fresh berries or used to generate syrup, jam, desserts, and liquor [76]. Even before the arrival of the Spaniards in 1536, in Chile, indigenous people consumed this fruit and its leaves as an infusion, or a liquor named “murtado” [77]. In addition, aboriginal people have used leaf infusions as an astringent in cases of diarrhea and dysentery [78]. Furthermore, roasted seeds can be used as a coffee substitute and, like dried fruits, they can be consumed in herbal infusions or food preparations [79,80]. Additionally, murta has also been used to treat infections and relieve pain. The chemical composition of *U. molinae* (leaves and fruit) presents a diversity of phenolic compounds and triterpenoids. In addition, it has been demonstrated that ethanolic extracts from *U. molinae* leaves have antioxidant capacity in vitro [7]. Studies on in vivo models through topical anti-inflammatory activity were assessed in a mouse ear model, inducing edema with either arachidonic acid or 12-O-tetradecanoylphorbol-13-acetate (TPA) [8]. Hexane, dichloromethane, crude ethyl acetate, and methanol extracts of *U. molinae* leaves showed intense anti-inflammatory activity against TPA, similar to the effect of indomethacin [9]. The same study has also described that the anti-inflammatory activities were mainly due to several pentacyclic triterpene acids, including 2-α-hydroxy derivatives alphitolic, asiatic, and corosolic acids [9]. Moreover, Goity et al. (2013) [10] demonstrated that derivatives of alphitolic acid and successive EtOAc and ethanol extracts from leaves showed anti-inflammatory activity, demonstrating that triterpenoids can be responsible in part of the anti-inflammatory activity, including madecassic and maslinic acids [10].

### 2.2. Buddleja globosa Hoppe (matico)

Due to their known wound and gastric ulcer healing effects, the leaves of *Buddleja globosa Hoppe* (Figure 3), known as matico, are widely used in traditional indigenous medicines. The ethnomedical use of matico is common in Chile, Argentina, Peru, and Bolivia and dates back to pre-Columbian times [12]. In Chile, in the Mapuche (known as *pañil*) and Aymara (known as *palquin*) cultures, this plant was used mainly as a healing agent and wound healer. In popular medicine, its indication for use is varied, from healing wounds (internal and external) and ulcers of all kinds, by applying poultices of leaves and stems or infusing their leaves to treat liver and gallbladder pain, burning or internal ailments, dysentery, stomach ulcers, scabies, and syphilis [81,82,83]. Studies regarding this extract have shown the involvement of saponins, sesquiterpenes, triterpenes, diterpenes, phenylethanoids, and flavonoids as principal constituents [12]. Wound healing and anti-inflammatory evaluations in vitro have been reported using a hydroalcoholic extract of the aerial parts and a dichloromethane root extract [13,14]. This study reported that a mixture of triterpenoids composed of α-amyrin (43.7%), β-amyrin (24.9%), and bauerenol (31.4%) was responsible for the anti-inflammatory activity observed.

### 2.3. Schinus polygamus (Cav.) Cabrera (huingán)

*Schinus polygamus* (Cav.) Cabrera (vernacular name huingán) is a tree of about 1.2–2 m in height (Figure 4). In Chile, it grows from the region of Arica and Parinacota (north of Chile) to the region of Araucanía (central Chile) [84]. Additionally, it is a shrub cultivated in Egypt for ornamental purposes [67]. According to folk medicine, the infusion of its leaves has been used for wound cleansing, and its bark decoction produces a balsamic essence used to treat arthritis. The latex that emanates from the bark is used as a plaster for muscle and tendon pain, dislocations, fractures, and skin irritations. The resin is recommended to treat chronic bronchitis [85,86]. Its fruit infusion has antipyretic, analgesic, and anti-inflammatory properties for arthritic pain and an anti-microbial effect for wound cleansing. Phytochemically, *S. polygamus* is a rich source of flavonoids, bioflavonoids, tannins, anthocyanins, phenolic acids, sterols, triterpenes, and volatile oils [87]. *S. polygamus* have different biological activities such as antipyretic, anti-inflammatory, analgesic, antioxidant, hepatoprotective, and anti-microbial activities [15]. Currently, essential oils have been used for centuries in traditional medicine. In addition, they occupy a prominent position in different industrial purposes, mainly in perfumes, pharmaceutical formulations, and food as flavoring and preservatives [85]. Furthermore, another study of its analgesic and anti-inflammatory properties described the isolation of β-sitosterol, one of the major secondary metabolites, and α- and β-pinene terpenoids, which are significant constituents of essential oil. This finding could explain the anti-inflammatory and analgesic activity with its resultant pain treatment [88].

### 2.4. Quillaja saponaria Mol. (quillay)

*Quillaja saponaria* Mol. is a tree of the Coastal Range and the foothills of the Andes in semiarid central Chile (Figure 5) [89]. Its inner bark has long been used for hair and wool washing. Moreover, the indigenous Mapuche people have used it to relieve toothache and treat inflammation, especially in the respiratory system [90]. Moreover, this tree has been used since pre-colonial times as a detergent. Its raw saponins are used as a foaming and emulsifying agent [91]. The main uses of *Quillaja saponaria* extracts are as emulsifiers in cosmetics, food, and beverages, as vaccine adjuvants, and as a biocide [92]. *Q. saponaria* extracts involve a complex mixture of glycosides and sugar esters, mainly of the pentacyclic triterpene quillaic acid [54]. The extracts have been successfully used in animals and humans as an antifungal, antibacterial, antiviral agent, and vaccine coadjuvant [93]. The anti-inflammatory activity of several of these compounds has been reported. For instance, the topical anti-inflammatory activity was evaluated using arachidonic acid (AA) and 12-O-tetradecanoyl phorbol-13 acetate (TPA), which induced inflammation in a mouse ear assay [23]. TPA acts primarily as a protein kinase C and NF-kB activator, promoting the expression of pro-inflammatory factors. On the other hand, AA presumably acts as a precursor of inflammatory mediators such as prostaglandins and leukotrienes. This study demonstrated that quillaic acid is a highly effective inhibitor of in vivo inflammation induced by topical application of either TPA or AA. Its apparent dual mode of action is unusual and may prove to be of some clinical relevance [23]. Some derivatives of quillaic acid are also active, although generally with a bias towards TPA- rather than AA-induced inflammation [23]. Moreover, another study demonstrated that quillaic acid, its methyl ester, and one of the oxidized derivatives of the latter elicited dose-dependent antinociceptive effects in two murine thermal models [24].

### 2.5. Acaena magellanica (Lam.) Vahl (cadillo)

*Acaena magellanica* (Lam.) Vahl, known as cadillo, is widely distributed in Argentina and Chile from sea level to 4000 m above sea level (Figure 6). The Yagan culture has used the infusion of this plant for pain, gallbladder, and allergies (the Yagan population were canoeing people who formerly lived in the area around the channels and southwestern coast of Tierra del Fuego in Chile and Argentina) [43]. Therefore, the anti-inflammatory activities of *A. magellanica* ethanolic, dichloromethane and methanolic extracts have been assessed in a guinea pig model. As expected, the compounds isolated from dichloromethane extracts were triterpenes and steroids with known anti-inflammatory effects [38].

### 2.6. Berberis microphylla (calafate)

*Berberis microphylla*, commonly called Magellan barberry (Figure 7), also known as calafate in Spanish, is an evergreen shrub [75]. Calafate is native to southern Argentina and Chile and is a Patagonian symbol [94,95]. Its edible blue-black berries are harvested for jams but are also eaten fresh. The Mapuches used the calafate fruit not only for its pleasant flavor but also for its many properties. The calafate fruit has been used to dye fabrics and to prepare drinks and remedies. Fruits from both *Berberis* species are collected and consumed by the Kawésqar, an indigenous people who live in Chilean Patagonia. There is a legend concerning this fruit in Patagonia, which says that whoever eats calafate fruits will not return to Patagonia and will be part of a spell [94,95]. Calafate is grown commercially for its fruit and potential medical uses [94,95]. A decoction of the bark brings down a fever, whereas fruits are used to combat diarrhea [96]. Some foliar extracts showed an excellent sun protection factor [97]. The use of this native fruit is gaining international interest, which is mainly driven by its high content of polyphenols [98,99]. Polyphenols are metabolites with well-known positive health effects [100]. Anthocyanins, the main polyphenolic compounds, have been reported to possess antioxidant and anti-inflammatory features [101,102]. It has been identified that the native Chilean fruits maqui and calafate present a high content of anthocyanins [56]. The shrub produces dark-skinned barberry with high polyphenol content and antioxidant capacity, which are typically consumed fresh or as juices, marmalade, and infusions. Anthocyanins, hydroxycinnamic acids, and flavonols are the main phenolic compounds described in the fruit [103]. It has a potent antioxidant capacity similar to other native berries, which correlates with the fruit’s high content of total polyphenols and its concentration of anthocyanins. The consumption of these compounds has been proposed as a method of protection against diseases [16,97]. The ability of these fruit extracts to modulate the inflammatory response of an in vitro adipocyte–macrophage interaction has been evaluated, showing that they possess essential anti-inflammatory and antioxidant features [16]. It was demonstrated that all evaluated extracts significantly prevented LPS-induced NO secretion by macrophages; however, the calafate extract induced a more drastic effect than other berries [16]. Additionally, in vitro studies highlighted that it could modify the inflammatory response produced by the interaction between adipocytes and macrophages, which correlates with the high content of total polyphenols in the fruit and the concentration of anthocyanins [104,105]. Moreover, the relative change of iNOS and TNF-α and IL-10 mRNA expression was studied to evaluate whether these extracts would modulate the gene expression of cytokines associated with specific inflammatory pathways. A protective effect was observed only with maqui and calafate extracts, compared to other berry extracts, regarding iNOS expression. A similar pattern was observed in the TNF-α gene expression. Finally, all the extracts significantly prevented LPS-induced IL-10 secretion [16].

### 2.7. Aristotelia chilensis (Mol.) Stuntz (maqui)

*Aristotelia chilensis* (Mol.) Stuntz, popularly known as maqui, also known as maquei, queldrón, queldón, clone, coclon, koelon, and Chilean blackberry (Figure 8), is a 4–6 m evergreen tree with yellow flowers and edible, black-colored fruit, which grows in central and southern Chile and southwestern Argentina [75,106]. For the Mapuche people, maqui is one of the sacred plants, a symbol of peaceful intention and goodwill [107]. The Spanish conquerors described maqui as a medicinal plant to treat sore throats, intestinal tumors, diarrhea, fever, or wounds [53,86,108,109]. Chilean folk medicine attributes various properties to the leaves, such as astringent and febrifuge properties, anti-diarrheal, anti-inflammatory, analgesic, anti-hemorrhagic, antioxidant, and cardio-protective activity [18,19,41,53,74]. Additionally, Agulló et al. 2021 showed the antinociceptive effects of maqui berry using a nociceptive pain model (formalin test) in mice [98]. Chamorro & Ladio, 2020 [96] wrote about the increasing interest in maqui berries and their use as exotic food and a source of raw material in the food and pharmaceutical industries [110,111]. Pharmacologically, fruits and leaves of maqui have been investigated for anti-inflammatory, analgesic, antioxidant, anti-diabetic, antiviral, and anti-microbial activities [112]. Studies have demonstrated that *A. chilensis* aqueous extract exhibits anti-inflammatory effects and immunomodulatory activity related to atopic-like dermatitis [113]. Several epidemiological studies on human health have underlined the beneficial role of phenolic compounds in preventing arteriosclerosis, cardiovascular diseases, arthritis, diabetes, neurodegenerative diseases, and cancers [113,114]. The berries have increased uses and are broadly selected to develop healthy or potential functional foods because of their biological attributes [110]. Most of the biological actions described for maqui are related to the high content of phenols in their fruits [112]. Furthermore, it has been described that this species has a high concentration of anthocyanin pigments, giving its berries a characteristic dark violet color [53]. The maqui fruit has four times more antioxidant properties than other berries [114]. *A. chilensis* is rich in poly-glycosylated derivatives with high antioxidant capacity, which suggests antiatherogenic properties. Topical anti-inflammatory effects in TPA and arachidonic acid assays and analgesic activity observed when the dichloromethane fraction was administrated might be caused by the mixture of the pentacyclic triterpenoids ursolic acid and friedelin, together with quercetin 5,3-dimethyl ether [18]. It has been described that this flavonoid has a more significant anti-inflammatory effect than mefenamic acid [18]. Moreover, reports have suggested that the topical anti-inflammatory activity of plant extracts is due to the presence of these compounds, mainly to the high content of ursolic acid [73]. In addition, reports have proved the antioxidant properties of ursolic acid through a series of in vitro tests, such as radical scavenging assays [115]. Quercetin 3-O-β-D-glucoside and kaempferol in methanol fraction may inhibit local TPA-induced inflammation and analgesic activity. In vivo assays show that kaempferol has a significant dose-dependent anti-inflammatory and analgesic effect [17]. In addition to this anti-inflammatory effect, kaempferol shows apparent antioxidant activity. Thus, Nagao et al. 1999, demonstrated that kaempferol might suppress the in vivo formation of reactive oxygen species (ROS) and urate by inhibiting xanthine oxidase activity [20]. Caffeic and ferulic acids found in the infusion fraction could be responsible for the analgesic effect and the topical arachidonic acid-induced inflammation [21,22].

### 2.8. Fragaria chiloensis spp. Chiloensis (Frutilla blanca)

The native white Chilean strawberry (*Fragaria chiloensis* spp. *chiloensis*) (Figure 9) was exported to Chile from Europe in the early eighteenth century, and it is the maternal progenitor of the commercial strawberry (*Fragaria* × *ananassa*) [55]. Mapuche and Picunche people cultivated this plant, and the fruit was consumed as a nutritive food or fermented drink in ceremonial rites [116]. In this regard, flavonoids, mainly anthocyanins, are strawberries’ main constituents of phenolic compounds [117]. Anthocyanins are *widely known* as pigments with antioxidant and anti-inflammatory properties [32]. Notably, the nutritional composition of the strawberry varies considerably with its genetic background. White Chilean strawberries are a good source of phenolic antioxidants [118]. Furthermore, in vitro studies have shown that white Chilean strawberry fruit has a high free radical scavenging effect [119]. Additionally, in vivo experiments with Sprague–Dawley rats showed that their dietary supplementation with the aqueous extract of the native Chilean white strawberry for ten days before a lipopolysaccharide (LPS) challenge at a dose of 4 g kg^−1^ day^−1^ diminished the induced damage in the liver [33]. LPS challenge increased inflammatory markers or wounds [53,86,108,109], and oral administration of *F. chiloensis* before the LPS challenge was able to reduce the increment of serum cytokines to a comparable level to non-challenged animals. These data suggest that the native Chilean white strawberry has excellent potential to be used as a natural anti-inflammatory dietary supplement. Additionally, Chamorro et al. described antioxidant activity between Patagonian berries, *Berberis microphylla*, *Berberis darwinii*, and *Fragaria chiloensis* ssp., showing that the phenolic content and composition of the Argentinean Patagonia berries were similar to that reported for Chilean samples but with some chemical differences related to its polyphenols profile between eastern (Argentina) and western (Chile) Patagonia [120].

### 2.9. Eulychnia acida Phil., Cactaceae (copao)

*Eulychnia acida* Phil. is an endemic species of arid regions widely distributed in Coquimbo. This fruit has a long tradition in conventional Andean medicine (Figure 10). The traditional use of this fruit is by ingestion in a fresh form, which is exactly how people from the northern areas consumed it. This species belongs to the family *Cactaceae*, known to have significant antioxidant, antiproliferative and anti-inflammatory effects [34,35]. An investigation into the chemical constituents of the methanol extract of copao fruits showed that the phenolic-enriched extract had ferulic acid, 9,10-dihydroxy-4,7-megastigmadien-3-one hexoside, isorhamnetin, and quercetin glycosides [121]. In this regard, a recent study demonstrated the anti-inflammatory activity of the phenolic compounds of copao fruit extracts by measuring the in vitro ability to inhibit the pro-inflammatory enzymes lipoxygenase (LOX) and cyclooxygenases (COX-1 and COX-2) [34]. These results evidence the possible beneficial health effects of this native Chilean fruit.

### 2.10. Haplopappus remyanus (bailahuén)

Bailahuen is a native medicinal plant traditionally consumed as infused water (Figure 11). This plant is widely used, from Mapuche communities in the south to Aymara communities in the north of Chile [41]. The name bailahuén is derived from the Mapuche language and means boiling medicine [52]. It is well known to aid digestion, to have antispasmodic and antiseptic properties, and to improve liver function (choleretic and cholagogue properties) [122]. The natural habitat of this species is limited to the mountainous areas of Chile from the Atacama and Coquimbo regions. Nevertheless, in other regions of Chile, other endemic *Haplopappus* species, such as *H. multifolius* Phil., *H. remyanus* Reiche, and *H. taeda* Wedd, are all known locally as Bailahuen [39,122]. Regarding *H. remyanus* anti-inflammatory activity, it has been reported that a dichloromethane extract from fresh leaves exhibits a moderate topical anti-inflammatory effect. The chemical characterization of the studied species showed high levels of flavonoids and coumarins [39].

### 2.11. Geoffroea decorticans (chañar)

The tree *Geoffroea decorticans* Burk (*Fabaceae*), known as chañar (Figure 12), is a native species found in the forests of the Gran Chaco region in Argentina, in the Paraguayan and Bolivian Chaco, and northern Chile [123]. This tree has been used as food since ancient times to prepare liqueurs, jams, and even ice cream. The chañar fruit and its derivatives (syrup and flour) were essential to travelers crossing the Atacama Desert between the 19th and 20th centuries because they supplied abundant sugars through their fleshy pulp. Studies reported that, during 1787, in the Atacama zone, there were vast forests of “chañares”, which were protected due to their fruits, from which was made a drink called “quilapanada”, consumed during the festivals [124]. Farmers have built heritage and identity around the chañar, which is fundamentally manifested collectively in families through celebrations of ancestral cosmovision and in social celebrations such as birth [125,126,127]. Chañar fruit contains high amounts of sugar, fiber, and a complex mixture of polyphenols that grant medicinal properties such as antinociceptive action and antioxidant activity [26,27]. In addition, a recent study showed that the *G. decorticans* polyphenolic extract exhibits antioxidant activity, inhibiting pro-inflammatory enzymes, such as cyclooxygenase, lipoxygenase, and phospholipase A2 [28]. Therefore, the evidence supports the concept that chañar fruit flour may be considered a functional food with preventive properties against diseases associated with oxidative stress and inflammatory mediators [128].

### 2.12. Laretia acaulis (yareta or llareta)

*Laretia acaulis,* known in Chile as yareta, is a yellowish green (Figure 13), compact resinous cushion shrub grown in the high Andes in northeastern Chile. In the last two centuries, this shrub has been highly exploited, e.g., as fuel for mining copper and nitrate, although its extraction began to be regulated in 1941 [129]. Whole plant infusion is used in folk medicine as a gastric stimulus, diuretic, and analgesic, as well as a treatment for the common cold, migraine, neuralgia, pneumonia, rheumatism, and diabetes treatment [130]. Petroleum ether extract from the aerial part of *L. acaulis* (Cav.) Gill et Hook (Umbelliferae) contains a mulinane diterpenoid, 13-epimulinolic acid, two mulinane-type diterpenoids, mulinolic acid, and mulin-11,13-dien-20-oic acid [131]. Further investigations were performed using the same extract, which identified the presence of azorellane-type diterpenoids, azorellanol, and a new diterpenoid, named 7-deacetylazorellanol [132]. A third diterpenoid was also identified in *L. acaulis* extract, 13-epiazorellanol [36]. Research regarding the activity of the different compounds isolated from *L. acaulis* extracts determined that azorellanol exhibited the highest topical anti-inflammatory activity [36]. In addition, Borquez et al. showed that azorellanol and 7-deacetylazorellanol have potent anti-NF-κB activity [133]. Transcription factor NF-κB plays a crucial role in the inducible expression of genes mediating pro-inflammatory effects [134]. Therefore, inhibitors of NF-κB activity could potentially be developed as anti-inflammatory drugs. Moreover, antituberculosis activity has been reported based on diterpenoids isolated from *L. acaulis* [37].

### 2.13. Peumus boldus (boldo)

*Peumus boldus* Molina, commonly known as boldo (Figure 14), belongs to the family of *Monimiaceae*. Boldo is a native tree that grows abundantly in the more humid ecosystems of the Mediterranean climatic region of central Chile [135]. The boldo is surrounded by magic rites. For the ancient Mapuche culture, it was a plant where, under its shadow, witches were transformed into frogs, lizards, or snakes. These animals were feared because they were associated with evil or thought to announce a death. In addition, the “traditional authority” heading the spiritual ceremonies was usually carved into the boldo’s trunk. Nowadays, some people still have these beliefs and spread boldo leaves around the house to avoid evil [136]. Boldo leaves have been used in folk medicine to treat headaches, earache, rheumatism, dropsy, dyspepsia, menstrual pain, and urinary tract inflammation [137]. Its leaves contain alkaloids, flavonoids, resin, tannins, and essential oil [138]. Boldo’s principal alkaloid active component is boldine, which is well known for its antioxidant effects. In addition, boldine has been shown to prevent oxidative stress-related pharmacological effects, including anti-inflammatory, antipyretic, antitumor-promoting, and antiatherogenic effects [29,30,31]. Remarkably, boldine has been reported to exert a neuroprotective effect through its anti-inflammatory properties [139].

## 3. Discussion

This review has shown the presence of molecules with anti-inflammatory properties present in the extracts from native Chilean plants. Most of the studies mentioned above have concluded their research by mentioning that the anti-inflammatory activity may be due to the inhibition of the enzyme cyclooxygenase, leading to the inhibition of prostaglandin synthesis. However, more extensive research must be conducted to determine the precise active mechanisms of action of these plant extracts and isolated compounds. Thus, it is necessary to study their properties and active mechanisms for future applications in treating common inflammatory diseases. Despite the high number of reported studies, more cellular and molecular studies should be carried out to increase our knowledge about these plants. Furthermore, most phytochemical studies have examined the most polarized fractions, where the most significant effect has usually been detected [140,141]. 

Consequently, a complete metabolic profile that includes polar and non-polar compounds is lacking; in addition, the biological effects of the extracts were mainly investigated in vitro. However, people usually consume these extracts as a regular infusion. In this context, some questions remain unsolved, such as “can an infusion successfully extract the required metabolites with their attendant anti-inflammatory properties?” Several of these studies have demonstrated that the flavonoids and the phenolic content of native Chilean plants can contribute to the antioxidant effects found in their extracts. Subsequently, natural compound-based antioxidant substances perform a defensive role in protecting against harmful free radical generation. Moreover, because they have antioxidant effects, flavonoids and phenolic compounds also exert a potent role as anti-inflammatory factors.

Several studies have reported that bioactive extracts and their natural compounds exert their anti-inflammatory properties by blocking two major signaling pathways, such as NF-κB and mitogen-activated protein kinases (MAPKs), which have the central role in the pro-inflammatory mediator’s production [142,143]. On the other hand, natural extracts are a natural, safe, and innovative therapy that can effectively treat allergic rhinitis [144]. Many cytokines are involved in this process, i.e., IL-9 could control the critical balance between homeostasis and inflammation. Furthermore, IL-9 signaling in the mucosal immune system is mainly present in allergic asthma, inflammatory bowel diseases, and other conditions [145]. Figure 15 illustrates the anti-inflammatory mechanisms related to the various phytochemicals shown.

Therefore, biochemical results based on in-vitro analysis demonstrate the potential role of various native Chilean plant extracts in inhibiting pro-inflammatory mediators. In this regard, it is expected that this review will help current and future researchers as they investigate the range of anti-inflammatory Chilean medicinal plants in which they can isolate active constituents. Furthermore, this research work may reveal newly-discovered molecules that will help fight against inflammatory diseases. Unfortunately, very few clinical scientific studies are reported supporting the traditional ancestral use of these plants in patients. [11,25] An interesting example is the progress in using quillaia saponins as a vaccine adjuvant [25]. Another recent example is the use of biofilms based on matico extracts in the treatment of wounds [11].

The medical systems of the Mapuche culture are composed of beliefs, knowledge, and practices. The primary responsibility for applying these functions is the machi, a healer whose election from the community originated due to different circumstances. The machi is in charge of treating diseases mainly attributed to natural and spiritual sources. The machi generally uses empiric (medicinal herbs) and some magic religious methods (praying, singing, and sounds) to increase the effectiveness of the treatment. Maqui is used as a soft drug that brings good luck, forgiveness, and recovery to the patient. These plants reconcile the patient with his or her family and community and help to overcome the trauma to help the patient return to his or her normal activities [146].

In a historical context, since the return to democracy in Chile in 1990, machis have prospered by healing modern life-related illnesses and the consequences of modernity through endemic medicinal plants. Mapuche medicine combines ancestral knowledge, nature, rituals, and spirituality and aims to address the root of an ailment, not just the symptoms. However, most machi are baptized Catholics and have re-signified elements of Catholicism, biomedicine, and national discourses in their healing epistemologies [147].

## 4. Materials and Methods

This review was performed based on the analysis of scientific data published about native Chilean plants and their health impacts. The information was gathered using the NCBI–Pubmed, Google Scholar, and Mendeley databases using the words “Chilean plant” and “anti-inflammatory properties/activities” (i.e., *Peumus boldus*, anti-inflammatory properties/activities). All scientific literature from 1965–2021 was used for this work. One hundred and nineteen total references were selected for this work.

## 5. Conclusions

In this review, we have introduced various native Chilean plant species whose anti-inflammatory properties have been analyzed. Nevertheless, it seems that studying active compounds is not sufficient. Chilean indigenous ancestral culture has a connection with nature that is fundamental for healing. Exploring ancestral village data shows that the therapeutic efficacy of this family medicine (that of the Mapuche people is the most well-known) is not only based on the aforementioned active compounds but also related to the symbolic meaning attributed by healers. Therefore, to fully understand these plants’ therapeutic efficacy, it is necessary to understand the sociocultural context in which they are used, in conjunction with an extensive study of their pharmacological properties. Thus, this approach could help to create a clearer vision of the healing effects of plant extracts.

The production and concentration of metabolites in plants depend on several factors, including environmental conditions that vary throughout Chile. The cultivation and harvesting conditions of the species also play a significant role in maintaining a good yield and quality biomass to prepare future pharmaceutical formulations. On the other hand, the stability of these herbal mixtures and phytochemicals must be reviewed in the production processes of phytopharmaceuticals. Proper storage will guarantee the stability of the active compounds extracted from plants, and we emphasize the need to carry out future studies on stability and storage in plant extracts for medicinal use.

## Figures and Tables

**Figure 1 pharmaceutics-15-00897-f001:**
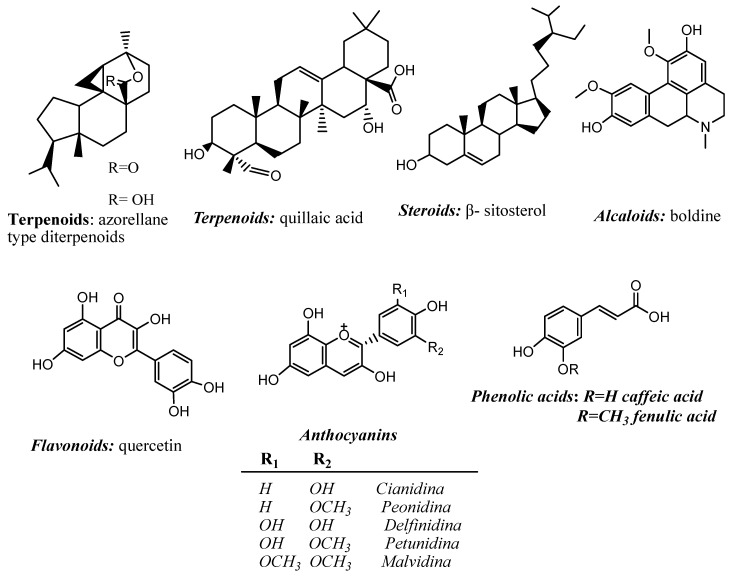
Some examples of genins found as secondary metabolites in native Chilean plants reported to have anti-inflammatory activity.

**Figure 2 pharmaceutics-15-00897-f002:**
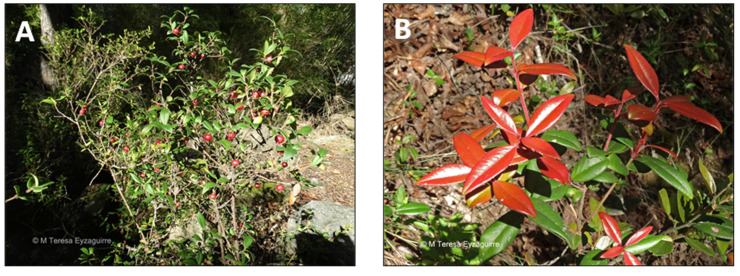
*Ugni molinae*. (**A**) Detail of the plant. (**B**) Leaves of the plant. (www.fundacionphilippi.cl) (accessed on 18 January 2023). https://fundacionphilippi.cl/catalogo/ (accessed on 18 January 2023).

**Figure 3 pharmaceutics-15-00897-f003:**
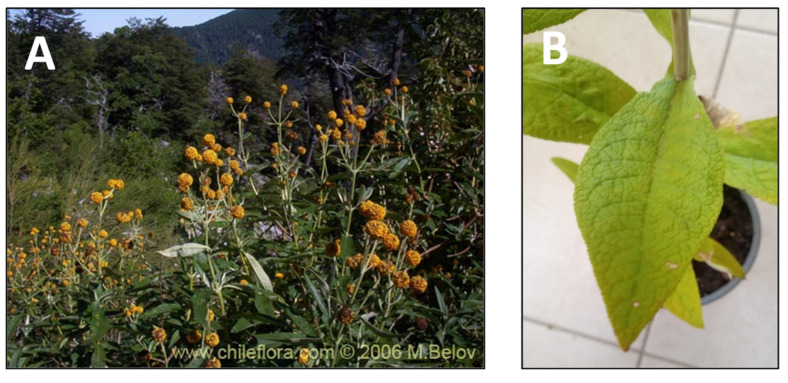
*Buddleja globosa.* (**A**) Detail of the plant. (**B**) Leaves of the plant. (www.chileflora.com) (accessed on 18 January 2023).

**Figure 4 pharmaceutics-15-00897-f004:**
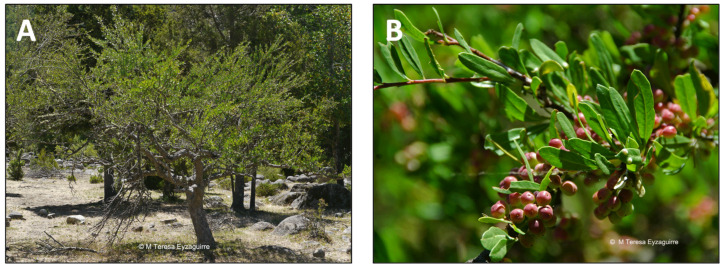
*Schinus polygamus*. (**A**) Detail of the tree. (**B**) Leaves of the tree. (www.fundacionphilippi.cl) (accessed on 18 January 2023).

**Figure 5 pharmaceutics-15-00897-f005:**
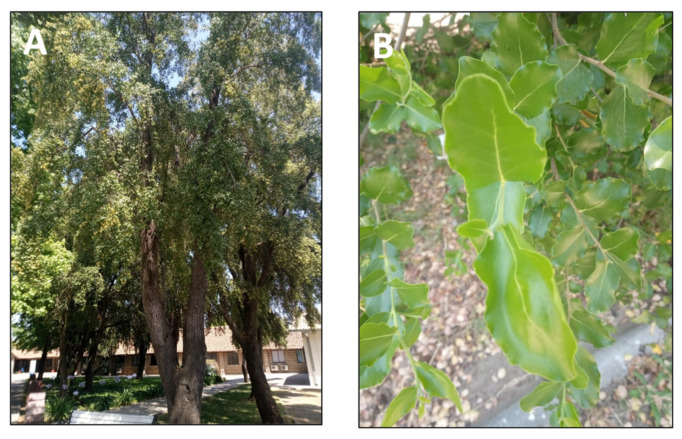
*Quillaja saponaria*. (**A**) Detail of the tree. (**B**) Leaves of the tree.

**Figure 6 pharmaceutics-15-00897-f006:**
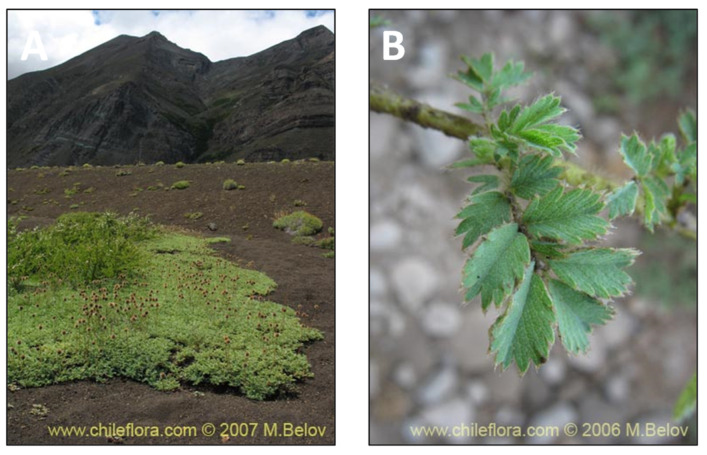
*Acaena magallánica.* (**A**) Detail of the plant. (**B**) Leaves of the plant. (www.chileflora.com) (accessed on 18 January 2023).

**Figure 7 pharmaceutics-15-00897-f007:**
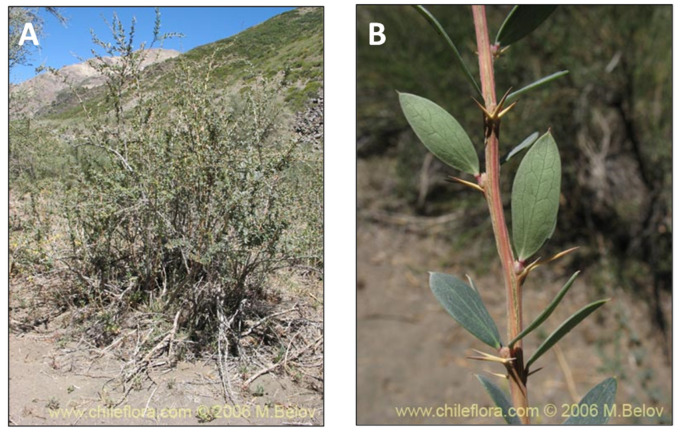
*Berberis microphylla.* (**A**) Detail of the plant. (**B**) Leaves of the plant. (www.chileflora.com) (accessed on 18 January 2023).

**Figure 8 pharmaceutics-15-00897-f008:**
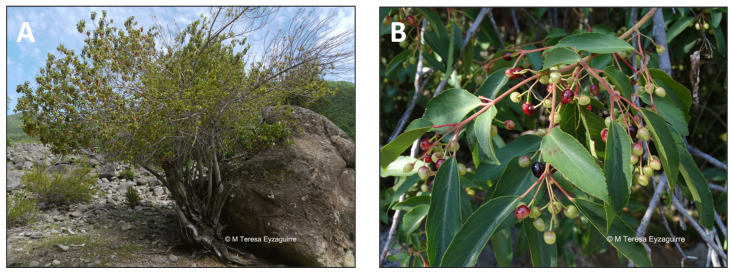
*Aristotelia chilensis.* (**A**) Detail of the tree. (**B**) Leaves of the plant. (www.fundacionphilippi.cl) (accessed on 18 January 2023).

**Figure 9 pharmaceutics-15-00897-f009:**
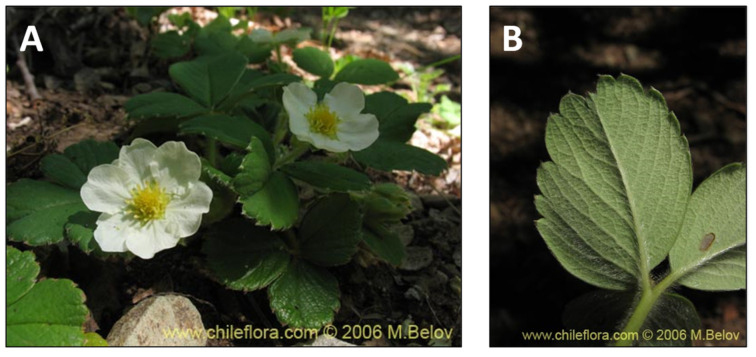
*Fragaria chiloensis.* (**A**) Detail of the tree. (**B**) Leaves of the plant. (www.chileflora.com) (accessed on 18 January 2023).

**Figure 10 pharmaceutics-15-00897-f010:**
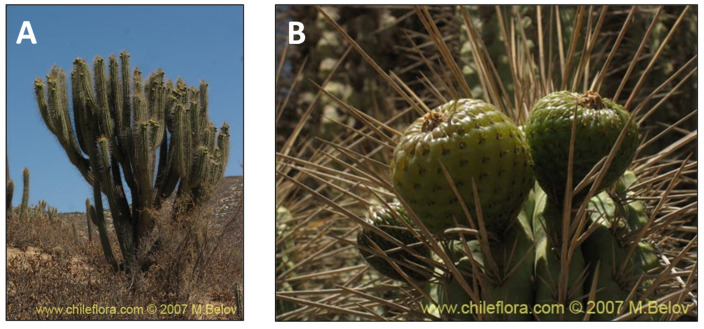
*Eulychnia acida.* (**A**) Detail of the tree. (**B**) Leaves of the fruit. (www.chileflora.com) (accessed on 18 January 2023).

**Figure 11 pharmaceutics-15-00897-f011:**
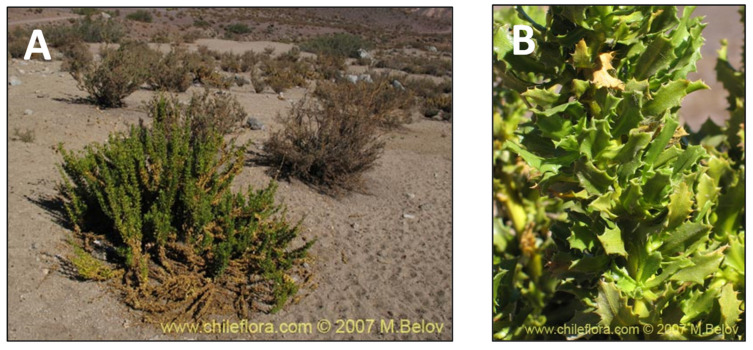
*Haplopappus remyanus.* (**A**) Detail of the tree. (**B**) Leaves of the plant. (www.chileflora.com) (accessed on 18 January 2023).

**Figure 12 pharmaceutics-15-00897-f012:**
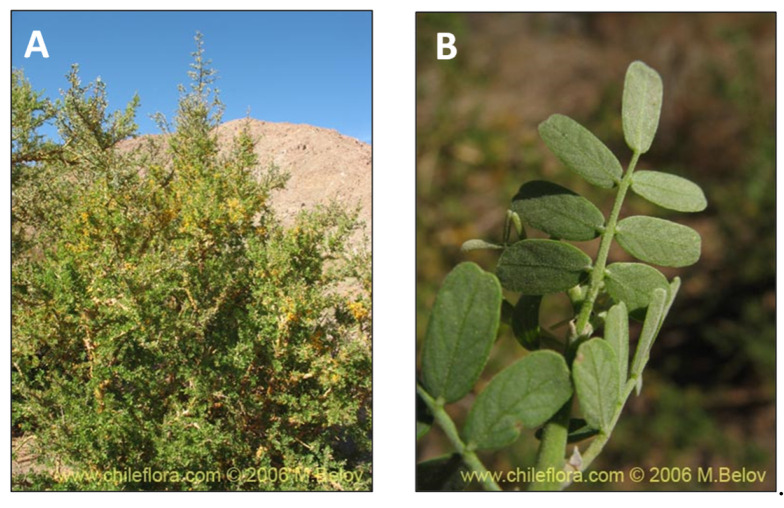
*Geoffroea decorticans.* (**A**) Detail of the tree. (**B**) Leaves of the plant. (www.chileflora.com) (accessed on 18 January 2023).

**Figure 13 pharmaceutics-15-00897-f013:**
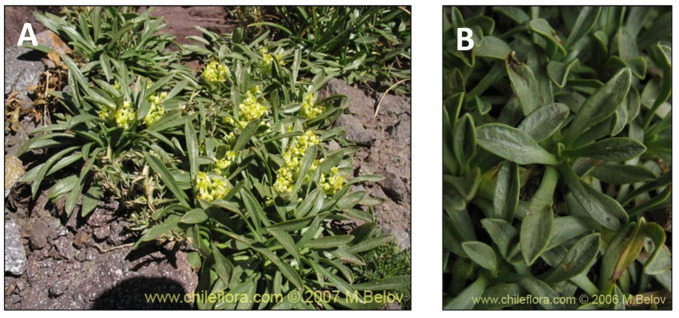
*Laretia acaulis.* (**A**) Detail of the plant. (**B**) Leaves of the plant. (www.chileflora.com) (accessed on 18 January 2023)..

**Figure 14 pharmaceutics-15-00897-f014:**
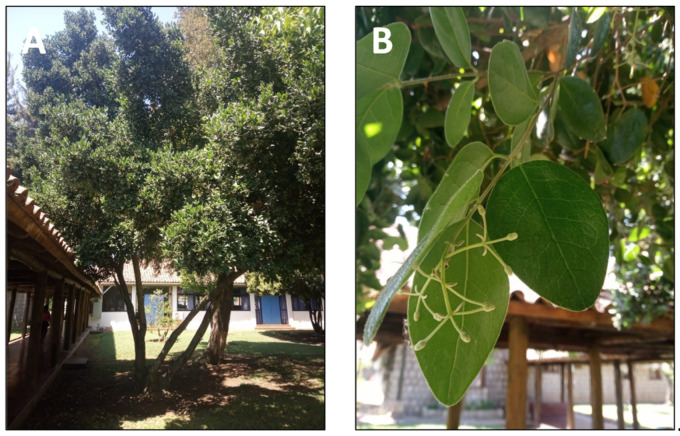
*Peumus boldus*. (**A**) Detail of the tree. (**B**) Leaves of the plant.

**Figure 15 pharmaceutics-15-00897-f015:**
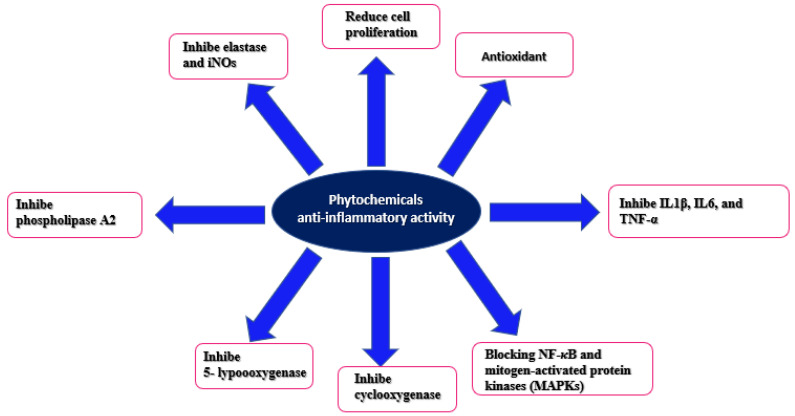
Anti-inflammatory mechanisms related to the various phytochemicals present in native Chilean plants.

**Table 1 pharmaceutics-15-00897-t001:** Plants species and its pharmacological studies.

Plant Species	Traditional Uses	Phytochemical Bioactives	In Vitro/In Vivo Studies	Clinical Studies
*Ugni molinae* Turcz	edible fruits, pain relief, diarrhea	phenolic compounds, pentacyclic triterpene acids such as corosolic acid	[7,8,9,10]	[11]
*Buddleja globosa* Hoppe	healing wounds, dysentery, stomach ulcers, scabies, syphilis	saponins, sesquiterpenes, triterpenes, diterpenes, phenylethanoids, flavonoids	[12,13,14]	NR
*Schinus polygamus* (Cav.) Cabrera	wound cleansing, arthritis	α- and β-pinene terpenoids, essential oil	[15]	NR
*Berberis microphylla*	edible fruits, pain relief, diarrhea	polyphenols, anthocyanins	[16]	NR
*Aristotelia chilensis* (Mol.) Stuntz	astringent and febrifuge properties, anti-diarrheal, anti-inflammatory, analgesic, anti-hemorrhagic	flavonoids, caffeic and ferulic acids	[17,18,19,20,21,22]	NR
*Quillaja saponaria* Mol.	pain relief	pentacycle triterpenoids	[23,24]	[25]
*Geoffroea decorticans*	edible fruit	sugar, fiber, and a complex mixture of polyphenols	[26,27,28]	NR
*Peumus boldus* Mol.	headaches, rheumatism, dyspepsia, menstrual pain, urinary tract inflammation	alkaloids, flavonoids, essential oil	[29,30,31]	NR
*Fragaria Chiloensis spp. chiloensis*	edible fruit	polyphenols, anthocyanins	[32,33]	NR
*Eulychnia ACIDA* Phil	edible fruit	flavonoids	[34,35]	NR
*Laretia acaulis*	gastric stimulus, diuretics, analgesics, rheumatism, diabetes treatment	mulinane diterpenoid	[36,37]	NR
*Acaena magellanica* (Lam.) Vahl	pain, gallbladder, allergies	triterpenes and steroids	[38]	NR
*Haplopappus remyanus*	Antispasmodic, antiseptic	flavonoids and coumarins	[39]	NR

## Data Availability

Not applicable.

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
