# Peer review of "Anti-Inflammatory Chilean Endemic Plants"

_pharmaceutics, 2023, doi:10.3390/pharmaceutics15030897_

Round 1

Reviewer 1 Report

1. The article is not organized in a good manner. The following modifications are required.

a) Put up a photo of each plant and leaf.

b) Anti-inflammatory effects should be grouped for each plant in vitro, in vivo and clinical study.

c) Make a table for plants' parts responsible for the anti-inflammatory effect and mention the active metabolites present in that part. 

2. Table 1 is missing but I understand the content to be.

3. How IL-9 is associated with inflammation, can be mentioned in the introduction with the following article (https://doi.org/10.1007/978-1-4939-6877-0_3) cited.

Author Response

Dear reviewer, thank you for your advice.

1. The review was ordered. Images (photos) of each plant are included as you suggested (Figures 2 to 14).
2. Table 1 has been rearranged and includes what you request in point 1b.
3. Point 1c is included in Figure 1
4. Your suggestion has been included

Reviewer 2 Report

I have a few questions and some corrections. See the enclosed file, please.

Author Response

Dear reviewer, thank you for your advice.

All his suggestions were considered.

Reviewer 3 Report

The manuscript submitted by Otero et al. reviews the anti-inflammatory properties of phytochemicals from medicinal plants native to Chile. The introduction of various indigenous herbal plants and their history of cultural and traditional medicinal use in various cultures in Chile is informative. The manuscript may be accepted for publication after considering the below suggesions:

1)    Please write some specific limitations and side effects of the existing anti-inflammatory drugs, including steroidal and non-steroidal, and immuno-suppressants in the introduction part.

2)    The authors can provide one or two examples wherever a general introduction is given. For instance, the section “Native plants and their metabolites” is a broad general introduction about the abundance of phytochemicals and secondary metabolites present in plants their extraction and identification. The authors can provide some examples of specific phytochemicals with medicinal value and examples of some complex herbal mixtures from which the isolation and analysis of the bioactive compound are difficult.

3)    The authors can include one or two schemes to illustrate the anti-inflammatory mechanism of the various phytochemicals reviewed in this manuscript.

4)    The authors could display the chemical structure of some important phytochemicals that have high anti-inflammatory effect.

5)    The authors can discuss the efficacy of the traditional methods and modern extraction method if available. Also, suggest or recommend the need for comparative studies on the curing effect or anti-inflammatory effect of traditionally prepared medicine mixture and phytochemicals isolated by modern scientific methods, to obtain the best possible herbal medicine.

6)    Phytochemicals are sensitive to environmental conditions and their chemical nature may change with time. The authors can discuss the stability of these herbal mixture and phytochemicals in the review. Also, suggest the need for further study on stability and storage in the discussion or conclusion section.

7)    The authors could discuss some adverse effects of the herbal decoction prepared from the reviewed plant species, if any.

8)    The authors may provide some information about government rules and regulations for the preparation, distribution, and use of traditional herbal medicine or scientifically isolated phytochemicals in Chile.

Author Response

(The authors gave the same response as above.)

Round 2

Reviewer 1 Report

The authors have addressed properly the issues raised by reviewer.